# Gross anatomy, histology and blood vessel topography of the alimentary canal of the Inland Bearded Dragon (*Pogona vitticeps*)

**Elisabeth Engelke**[1]*, **Christiane Pfarrer**[1], **Katharina Radelof**[2], **Michael Fehr**[2], **Karina A. Mathes**[2]

**1** Institute for Anatomy, University of Veterinary Medicine Hannover, Hannover, Germany, **2** Clinic for Small Mammals, Reptiles and Birds, University of Veterinary Medicine Hannover, Hannover, Germany

* elisabeth.engelke@tiho-hannover.de

**Data Availability Statement:** All relevant data are within the manuscript.

**Funding:** This publication was supported by Deutsche Forschungsgemeinschaft (https://www.

## Abstract

Imaging techniques have proved to be crucial for diagnosis in reptile species. The topography of the internal organs of bearded dragons has been described in recent studies as meeting the small animal practitioners´ demand for knowledge concerning their anatomy. However, the nomenclature in the respective literature is not uniform, which could lead to misunderstandings concerning the respective and/or affected parts of the alimentary canal. Therefore, the aim of this study was to provide clear information on anatomy and histology of the alimentary canal of bearded dragons including supplying blood vessels. For the dissection of the alimentary canal, 11 Inland Bearded Dragons (*Pogona vitticeps*) were used (five males, six females), which had been euthanised for clinical reasons other than those concerning the digestive tract or had died spontaneously. The supplying arteries were demonstrated by injecting red latex into the aorta, while the intestinal veins were filled with blue latex via the portal vein. Microscopic examination was carried out on specimens of seven additional bearded dragons using routine histologic procedures. Macroscopically, the sections of the alimentary canal from oral to aboral were distinguished into oesophagus, stomach, small intestine, colic ampulla, colic isthmus, rectum and cloaca. Differentiation of the duodenum, jejunum and ileum was only possible when considering the bile duct, the vasculature and the histology of the organ wall. Arteries supplying the oesophagus and the final straight part of the large intestine originated from the aorta in a segmental manner. Between these, three unpaired arteries arose from the aorta. Their branches supplied stomach and intestine excluding its last part. Based on the findings of the present study, a nomenclature for the different parts of the alimentary canal and the supplying blood vessels of bearded dragons is suggested which is well understandable for veterinary practitioners and is based on zoological knowledge of reptiles.

dfg.de) and University of Veterinary Medicine Hannover, Foundation (https://www.tiho-hannover.de) within the funding programme Open Access Publishing. The funders had no role in study design, data collection and analysis, decision to publish, or preparation of the manuscript.

**Competing interests:** The authors have declared that no competing interests exist.

## Introduction

The Inland (or Central) Bearded Dragon (*Pogona vitticeps*) is an Australian agamid lizard species and is one of the most popular pet lizard species nowadays. These lizards are relatively easy to maintain and docile, so they make good pets [1]. Nevertheless, nearly 40 percent of lizard patients are affected by gastrointestinal disorders [2]. In some of these cases, e.g. foreign bodies, invagination or volvulus, a quick diagnosis is essential to save the animal's life. Thus, imaging techniques play an important role. However, they have their limitations as ultrasound displays the stomach and the large intestine reliably, but not the small intestine [3]. In another ultrasound study, the gastrointestinal tract could be investigated best when it was empty or filled with liquids, otherwise the signal was blocked by gas [4]. Radiography after administering contrast medium has proved to be safe and informative [5, 6], as it not only shows the whole alimentary canal but also proves its patency. The alimentary canal is the part of the digestive tract, which follows the pharynx and terminates with the anus or vent [7, 8], and cannot be examined via direct inspection.

To meet the increased demand for anatomical knowledge about the alimentary canal of the bearded dragon, several studies were published recently [3, 5, 6, 9]. However, the nomenclature used is not uniform; the stated reason for this is the absence of an official nomenclature [9].

Therefore, the aim of the present study was to suggest a nomenclature for the different parts of the alimentary canal and the supplying blood vessels of bearded dragons, which is both, easily understandable for veterinary practitioners and conform to the zoological terminology of reptiles.

## Material and methods

For the macroscopic examination, 11 adult bearded dragons were used (five males, six females), which had been euthanised for clinical reasons other than for those concerning the digestive tract, or had died spontaneously. The owners of the pets were asked for permission to use the cadavers for scientific purposes. The body length was measured from the tip of the snout to the vent (Snout Vent Length, SVL). The male lizards had an average SVL of 20.7 cm (+/- 2.36 cm, range from 17.0 to 23.0 cm) and the mean SVL of the six females was 18.5 cm (+/- 4.12, range from 11.8 to 23.5 cm).

Up until the examination, the cadavers had been stored at -18˚C. After thawing, the body cavity was opened by an incision in the median line of the ventral body wall. In order to demonstrate the blood vessels, coloured latex (60%; Wurfbain Nordmann, Hamburg, Germany; colour: Alpina Voll- und Abtönfarbe, Alpina Farben, Ober-Ramstadt, Germany) was injected for the arteries into the heart ventricle of 11 specimens, and additionally for the gastrointestinal veins into the portal vein of seven specimens. After the latex had hardened, the topography of the alimentary canal was observed in situ and photographs were taken (Lumix DMC-FZ150, Panasonic Corporation, Kadoma, Japan). Subsequently, the total alimentary canal was removed from the coelom and spread out to identify the different parts of the canal and the supplying blood vessels. The canal was opened along its antimesenterial side in order to inspect and describe the mucosal surface. The specimens were photographed for documentation purposes. In addition, exact measurements of the sections of the alimentary canal were taken and have been published recently [6].

For the histologic examination, eight additional bearded dragons (three males, five females), were used. Their body weight ranged from 60 g to 365 g. They had been euthanised for clinical reasons, which did not affect the digestive system. The alimentary canal was exenterated, the lumen carefully rinsed with tap water and the specimens were fixed in Lilly's

formalin (10%, neutrally buffered, Merck, Darmstadt, Germany) or in Bouin's solution. The entire alimentary canal was cut into one to three samples (around 10 mm in length) per segment, depending on the segments' length. The specimens were embedded in paraffin (Paraplast Bulk REF 36602012, Leica Mikrosysteme, Wetzlar, Germany), cut into 2–3 μm thick sections and stained with haematoxylin-eosin (HE) or Masson-Goldner trichrome.

The study was carried out in strict accordance with the recommendations from the Institutional Animal Care and Use Committee at the University of Veterinary Medicine Hanover, Germany, and was also approved by the Lower Saxony State Office for Consumer Protection and Food Safety, the regional office responsible for oversight of animal research (Protocol Number 33.9-42502-05-14A457).

## Results

### Gross anatomy

The **body cavity, coelom**, of the bearded dragon was lined by peritoneum, which was slightly pigmented in the cranial half of the body cavity, and black pigmented in the caudal half (Fig 1).

The caudal part of the pharynx was black pigmented and merged into the oesophagus at the cranial thoracic aperture. The pharynx was crossed ventrally by the trachea. Large arterial blood vessels, i.e. the aortic arches and the pulmonary arteries, were in close contact to the pharyngeal wall.

Following the pharynx, the **alimentary canal** began with the straight oesophagus (Fig 1A), which ran caudally between the two lungs (Fig 1L) and dorsal to the heart within its pericardium and the cranial parts of the liver (Fig 1K). Black pigmentation was also observed in the cranial part of the oesophagus and decreased gradually in the caudal direction until it was completely lost. Its caudal extent varied individually, but the pigmentation terminated always cranial to the stomach. The mucosa of the oesophagus showed thin longitudinal folds along its entire length (Fig 2A), which disappeared when the organ was stretched. The oesophagus merged into the cranial, cardiac part of the stomach without a macroscopically clearly visible border (Fig 1 $B_1$). The main part of the stomach, the body (Fig 1 $B_2$), was located paramedian on the left side of the coelom, where it ran caudolaterally. From its subsequent turn to the right, the pyloric part of the stomach (Fig 1 $B_3$) continued caudomedially. In some animals, it continued even craniolaterally on the right side of the body cavity. The surface structure of the gastric mucosa was smooth along its entire length (Fig 2B). The stomach had a dorsal and a ventral mesogastrium. The dorsal mesogastrium originated median at the dorsal body wall where the descending aorta was located. It contained the gastric arteries and was attached to the dorsal margin of the stomach. Where the stomach turned to the right, the attachment of the mesogastrium stayed dorsal, taking the short way along the smaller curvature to the pylorus (Fig 1$B_4$). The ventral mesogastrium connected the ventral gastric margin to the left lobe of the liver, forming the lesser omentum, and continued from the liver to the ventral body wall as the falciform ligament or ventral mesohepaticum. At the pyloric part, the lesser omentum attached cranially at the smaller curvature of the stomach. Thus, both mesogastria were located close together at the pylorus and the initial part of the duodenum. In this region, the lesser omentum contained the small, ventral lobe of the pancreas and the common bile duct as well as the portal vein in its caudoventral free border. Parallel to the caudal half of the stomach, the large dorsal lobe of the pancreas was located in the dorsal mesogastrium (Fig 1F). The body of the pancreas was located directly at the duodenum, opposite to the attachment of both mesogastria, connecting both pancreatic lobes. The dark coloured spleen (Fig 1G) could be found at the cranial end of the beige-coloured dorsal lobe of the pancreas. The stomach terminated

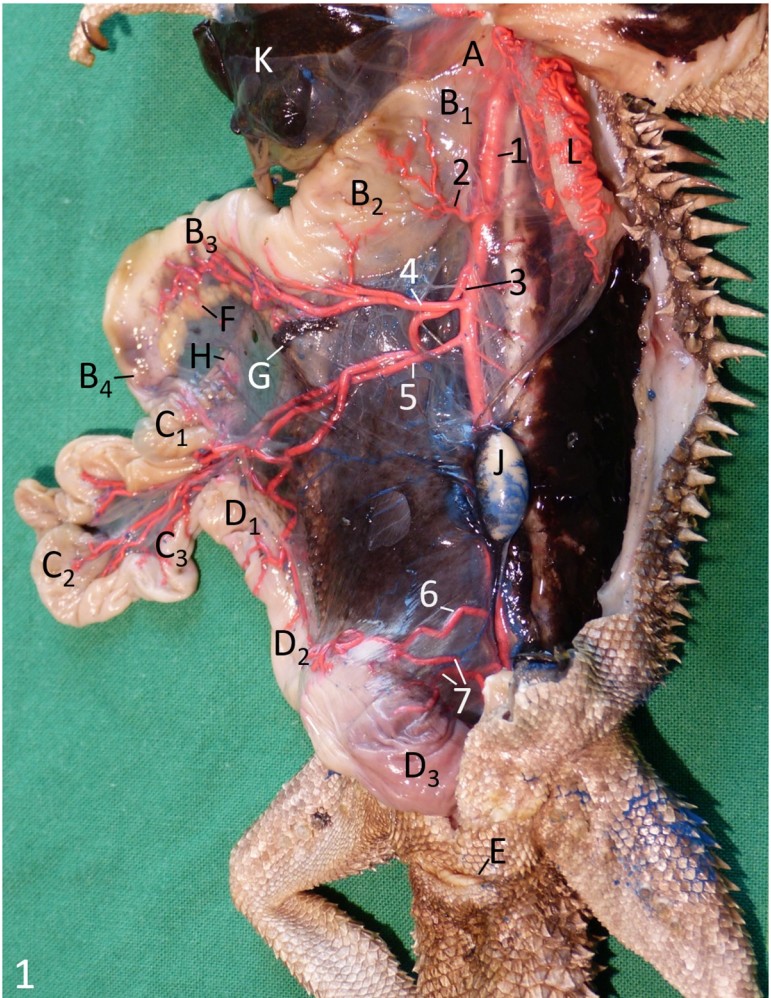

**Fig 1. Male Inland Bearded Dragon (*Pogona vitticeps*), ventral aspect, coelom opened, alimentary canal and liver displaced to the right, arteries filled with red latex.** A oesophagus; B1 cardiac part; B2 body of stomach; B3 pyloric part of stomach; B4 pylorus; C1 duodenum; C2 jejunum; C3 ileum; D1 colic ampulla; D2 colic isthmus; D3 rectum; E vent; F pancreas; G spleen; H bile duct; K liver; J left testis; L left lung. 1 descending aorta; 2 gastric artery; 3 ileocolic artery; 4 celiac artery; 5 cranial mesenteric artery; 6 caudal mesenteric artery; 7 rectal arteries.

with a clearly visible pylorus (Fig 1B₄), which was marked by palpable musculature and a circular mucosal fold (Fig 2B/a), and opened into the small intestine. The diameter of the small intestine (Fig 1C₁–1C₃) was much smaller than that of the oesophagus and stomach. The small intestine formed an irregular U-shaped loop and was suspended by a comparatively long mesentery and a mesenteric plica between the duodenum and the colic ampulla. The first part of the small intestine, the duodenum (Fig 1C₁), was joined by the clearly visible bile duct (Fig 1H). Its surface structure was characterised by tightly packed, comparatively high, and extremely undulating longitudinal folds (Fig 2C/b). The second part, the jejunum (Fig 1C₂), was considerably longer than the duodenum. Its mucosal surface had folds similar to those of the duodenum. However, they were straighter, and therefore arranged in a looser manner (Fig 2D/b). The third part, the ileum (Fig 1C₃), had a slightly smaller lumen than the jejunum, if the small intestine had been empty at the time of death. The ileal mucosa still showed numerous folds, but these had decreased remarkably in height and were straight longitudinally (Fig

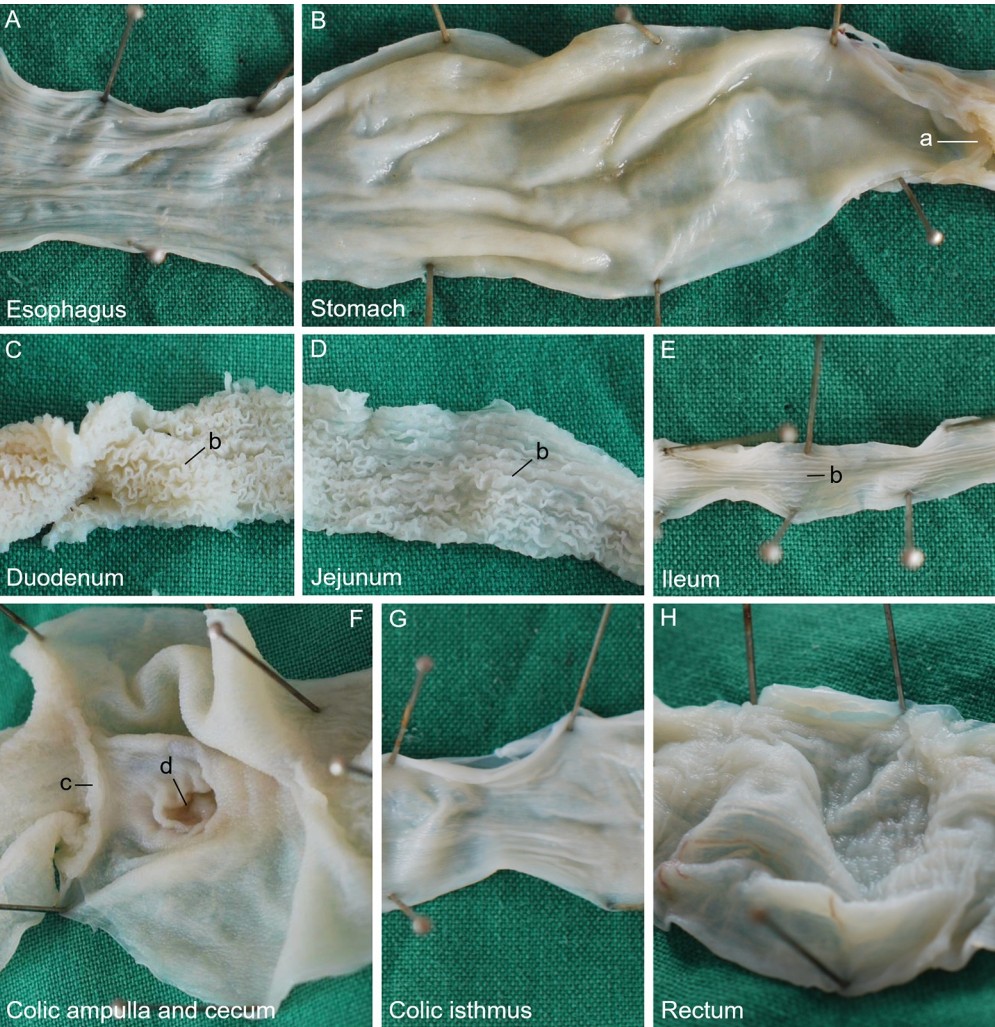

**Fig 2.** Surface structure of the alimentary canal of the Inland Bearded Dragon (*Pogona vitticeps*): (A) oesophagus, (B) stomach, (C) duodenum, (D) jejunum, (E) ileum, (F) colic ampulla and caecum, (G) colic isthmus and (H) rectum. a pylorus; b intestinal folds; c ileal orifice; d caecocolic orifice.

2E/b). There was a distinct border to the large intestine (Fig 1D$_1$–1D$_3$) because its first part was extremely widened, almost forming a sphere, the colic ampulla (Fig 1D$_1$). A circular mucosal fold marked the transition between the small and large intestine (Fig 2F/c). The mucosa of the colic ampulla lacked folds; instead, it showed small longitudinal protrusions (Fig 2F). At the dorsal mesenteric border of the colic ampulla, the opening (Fig 2F/d) to a 3 to 6 mm long, blind-ending diverticulum, the rudiment of the caecum, could be distinguished clearly in nine out of 11 animals. Opposite to the ileal orifice, the colon continued as a narrow and relatively short colic isthmus (Fig 1D$_2$), of which the mucosal surface showed a pattern of protrusions like in the colic ampulla. Additionally, small longitudinal folds were present (Fig 2G), which could be flattened when distending this intestinal part manually. The colic isthmus merged into the final, nearly straight part of the large intestine, the rectum (Fig 1D$_3$), which was remarkably wider again. The surface of the rectal mucosa showed very small protrusions (Fig 2H). Some predominantly longitudinal folds were present in these parts of the large intestine. In its caudal half, the rectum became gradually less voluminous before terminating in the

cloaca (Figs 3, 4/E). The border between rectum and cloaca was visible through one circular mucosal fold, in one animal two folds were observed. The cranial part of the cloaca, the coprodeum, was separated by a single circular fold from the urodeum. This part had no definite border to the final part, the proctodeum, which opened with the vent (Fig 1E), a transverse slit, on the ventral body surface at the base of the tail.

## Vasculature

Most **arteries** supplying the alimentary canal originated from the descending aorta (Fig 3/1). In bearded dragons, two aortic arches arose from the left ventricle of the heart. The left arch was slightly stronger than the right one. Both aortae were very close to the cranial part of the oesophagus and each released three to four small and very short (length: 2–4 mm) arteries to this organ part. At the level of the $3^{rd}$ to $4^{th}$ thoracic vertebra, both aortae merged, forming one descending aorta (Fig 3/1). After the junction, two to four pairs of short (4–10 mm long) arteries originated from the aorta, and supplied the remaining part of the oesophagus in a segmental manner (Fig 3/2). At the level of the $6^{th}$ to $7^{th}$ thoracic vertebra, one to two pairs of gastric arteries (Fig 3/3) reached the cardiac part of the stomach (Fig 3/$B_1$). The following major part of the alimentary canal was provided with three unpaired arteries leaving the aorta between the $8^{th}$ and the $11^{th}$ or $12^{th}$ thoracic vertebra, the ileocolic, celiac and cranial mesenteric arteries. In one animal, the origin of the first two major arteries was different in the way that the celiac artery left the aorta cranial to the ileocolic artery.

The first artery, ileocolic artery (Fig 3/4), originating at the level of the $8^{th}$ to $9^{th}$ thoracic vertebra ran caudally and split into two almost equivalent vessels, of which the cranial one supplied the duodenum (Fig 3/10) and the caudal one the ileum, colon and cranial part of the rectum (Fig 3/4'). The second artery, the celiac artery (Fig 3/5), originating one length of a vertebra caudal to the first one ($9^{th}$– $10^{th}$ thoracic vertebra), crossed the ileocolic artery on its left side cranially. In one animal, the origin of both major arteries (Fig 3/4, 5) was different: The celiac artery left the aorta cranial to the ileocolic artery. The celiac artery supplied the major part of the stomach (Fig 3/$B_2$–3/$B_4$), the dorsal lobe of the pancreas (Fig 3/F), the spleen (Fig 3/G) and the beginning of the duodenum (Fig 3/$C_1$). At first this artery gave off short gastric arteries, which ran through the dorsal mesogastrium before attaching itself to the dorsal margin of the stomach, while other short branches entered the spleen and the pancreas. The continuation of the celiac artery crossed the stomach on its right side to the ventral margin (Fig 4/12). Here, at the insertion of the ventral mesogastrium, it split into a cranial branch to both the body and the cardiac part of the stomach and a caudal smaller branch to the pyloric part and the duodenum. The origin of the third artery, the cranial mesenteric artery (Fig 3/6), was slightly more variable: It originated one to two vertebral lengths caudal to the second artery between the $10^{th}$ and $12^{th}$ thoracic vertebra. It crossed the caudal branch of the ileocolic artery on its right side and split into two equivalent vessels, which both ramified again to reach the small intestine, especially the jejunum, with several branches (Fig 3/11). The caudal mesenteric artery (Fig 3/7) left the aorta at the level of the $2^{nd}$ to $3^{rd}$ lumbar vertebra in order to supply the rectum. Before reaching the intestine, this artery bifurcated into a cranial and a caudal branch. The caudal branch ramified into relatively short branches to the rectum, whereas the cranial branch split up into some short branches to the cranial part of the rectum and a long branch, which ran further cranially to anastomose with the colic branch of the ileocolic artery. Between the caudal mesenteric artery and the final bifurcation of the aorta, two to three arteries arose from the aorta again in a segmental manner at the level of the last lumbar vertebrae ($3^{rd}$ to $5^{th}$). In one male animal, the last two rectal arteries seemed to be doubled, so that a total of five arteries was observed. In all animals, these arteries were unpaired, ramifying shortly

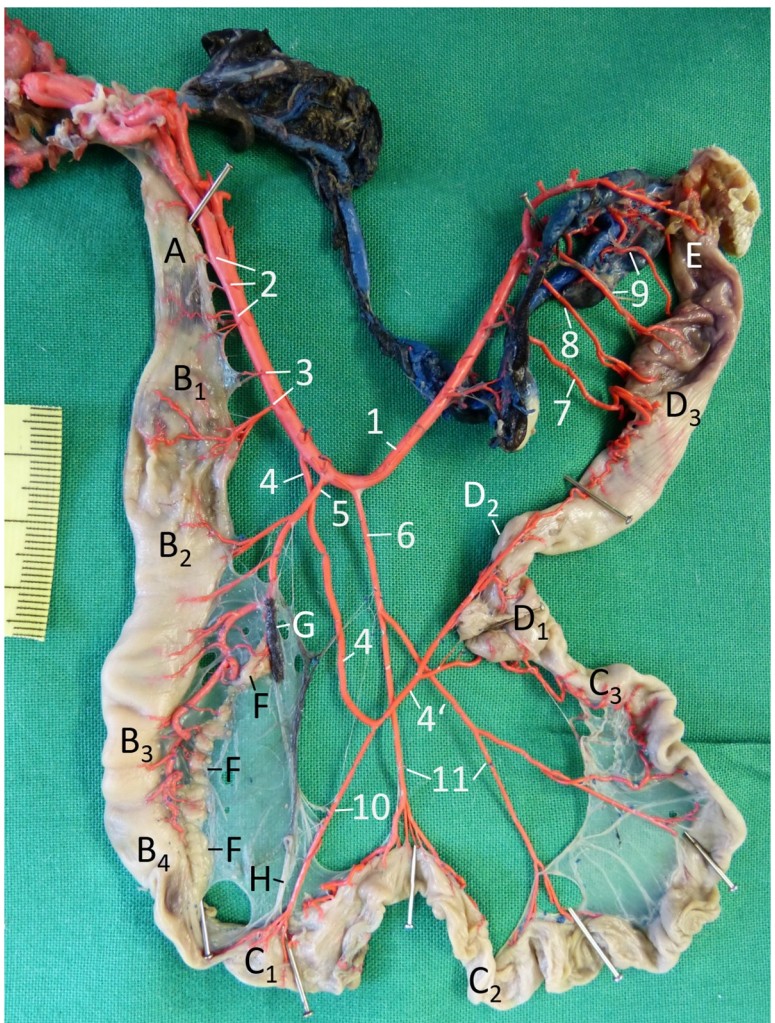

**Fig 3. Arteries of the alimentary canal of a male Inland Bearded Dragon (*Pogona vitticeps*), exenterated, fixed with pins to be displayed single-planed, left aspect; arteries filled with red latex.** A oesophagus; B1 cardiac part of stomach; B2 body of stomach; B3 pyloric part of stomach; B4 pylorus; C1 duodenum; C2 jejunum; C3 ileum; D1 colic ampulla; D2 colic isthmus; D3 rectum; E cloaca; F pancreas; G spleen; H bile duct. 1 descending aorta; 2 oesophageal arteries; 3 gastric arteries; 4, 4' ileocolic artery; 5 celiac artery; 6 cranial mesenteric artery; 7 caudal mesenteric artery; 8 rectal arteries; 9 cloacal arteries; 10 duodenal artery; 11 jejunal arteries.

before reaching the rectum (Fig 3/8). The last of these rectal arteries came from the aortic final bifurcation and reached the intestine at the junction between the rectum and cloaca. For the blood supply of the cloaca, arteries derived from the iliac and renal arteries on both sides (Fig 3/9).

When the entire dorsal mesentery—including the mesogastrium—was displayed single-planed (Fig 4), it became visible that in the mesentery all arteries of the alimentary canal crossed the respective veins on the right side—with one exception: the duodenal artery of the ileocolic artery. The majority of the veins were not parallel to the arteries of the alimentary canal (Fig 4).

The gastrointestinal **veins** draining blood from the stomach, as well as from the small and large intestine including the cloaca joined the portal vein (Fig 4/1), delivering the venous blood to the liver.

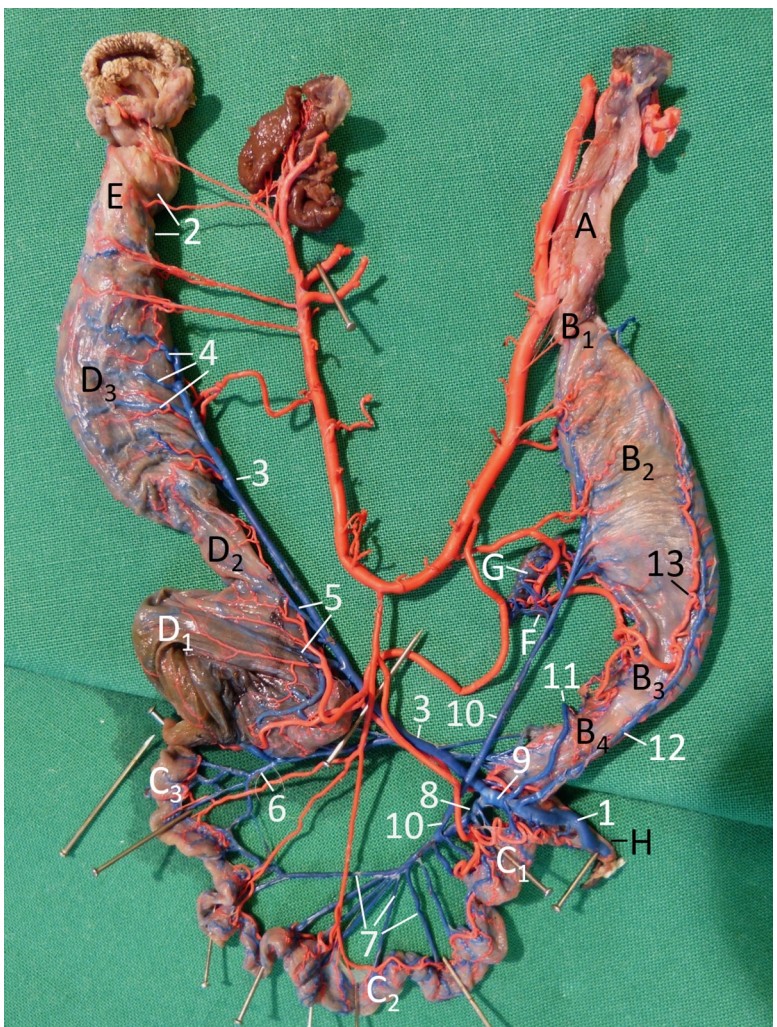

**Fig 4. Vascularisation of the alimentary canal of a female Inland Bearded Dragon (*Pogona vitticeps*), exenterated, fixed with pins to be displayed single-planed, right aspect; arteries filled with red latex, veins filled with blue latex: veins labelled.** A oesophagus; B1 cardiac part; B2 body of stomach; B3 pyloric part of stomach; B4 pylorus; C1 duodenum; C2 jejunum; C3 ileum; D1 colic ampulla; D2 colic isthmus; D3 rectum; E cloaca; F pancreas; G spleen; H bile duct. 1 portal vein; 2 ventral gastric vein; 3 right dorsal gastric vein; 4 common mesenteric vein; 5 cranial mesenteric vein; 6 caudal mesenteric vein; 7 left dorsal gastric vein, 8 jejunal veins; 9 ileal vein; 10 colic veins; 11 rectal veins; 12 ventral gastric artery.

The veins draining the blood from the cloaca (Fig 4/2) ran in the organ wall dorsally and subsequently cranially. Finally, the cloacal veins merged within the dorsal cloacal wall and formed the caudal termination of the caudal mesenteric vein (Fig 4/3). This vein ran cranially, leaving the organ wall in the caudal region of the rectum continuing into the rectal mesentery. Positioned directly dorsal to the organ, the caudal mesenteric vein was joined by several rectal veins (Fig 4/4) from both sides of the organ wall. At the colon, the distance between the intestine and the caudal mesenteric vein became greater, the latter receiving the colic veins (Fig 4/5): the caudal colic vein from the colic isthmus, and a little further cranially, two veins from the colic ampulla, the cranial colic veins, which met the caudal mesenteric vein at one point. A small venous branch from the proximodorsal colic wall and the caecal diverticulum joined the ileal vein.

The blood drained from the ileum was collected by small veins, which merged to form one ileal vein (Fig 4/6). In five out of seven animals, the ileal vein joined the cranial mesenteric vein, in the remaining two animals the caudal mesenteric vein. Four to six veins draining the jejunum, the jejunal veins (Fig 4/7), converged to become the cranial mesenteric vein (Fig 4/8). In the direction towards the duodenum, further (one to three) smaller and shorter jejunal veins joined the cranial mesenteric vein. Subsequently, the cranial mesenteric vein met the almost equally thick caudal mesenteric vein to form the common mesenteric vein (Fig 4/9). This latter vein was relatively short, because it became the portal vein (Fig 4/1) after receiving the major gastric vein(s). At the crossing of the portal vein and the duodenum, five to seven relatively small and short duodenal venous branches directly entered the portal vein. Additionally, one or two veins from the distal duodenum joined the cranial mesenteric vein, and some small veins from the proximal duodenum joined the gastric veins.

Three major veins drained blood from different parts of the stomach, two veins from the dorsal gastric margin and one vein from the ventral margin. The left vein from the dorsal margin collected blood from venous branches of the proximal two thirds of the stomach. In five out of seven animals, this left dorsal gastric vein (Fig 4/10) merged with the cranial mesenteric vein, in two animals it joined the caudal mesenteric vein. Independent of its termination, the left dorsal gastric vein drained the spleen (Fig 4/G) and dorsal lobe of the pancreas (Fig 4/F) on its way from the stomach. The right dorsal gastric vein (Fig 4/11) was slightly smaller; it drained blood from the distal third of the stomach and continued distally to the dorsal border of the duodenum. The large ventral gastric vein (Fig 4/12) received small veins from the entire ventral margin of the stomach beginning at the cardiac part. It ran caudally and crossed the pyloric part of the stomach to the dorsal border of the duodenum. Here, it joined the right dorsal gastric vein to form one very short vein that continued into the portal vein. In two out of seven animals, the right dorsal and the ventral gastric vein joined the portal vein separately, lying close together (Fig 4/11, 12).

## Histology

In general, the histology of the wall of the alimentary canal of the bearded dragon displayed four layers: mucosal tunic (Fig 5A–5D/1-1", Fig 6A–6C/1-1', Fig 7A–7D/1-1"), submucosal layer (Fig 5A–5D/2, Fig 6A–6C/2, Fig 7A–7D/2), muscular tunic (Fig 5A–5D/3-3', Fig 6A–6C/3-3', Fig 7A–7D/3-3') and serosal tunic (Fig 5A/4, Fig 6A, 6C/4, Fig 7A–7C/4) or adventitia (Fig 7D/4').

The **oesophagus** showed a pseudostratified or bilayered columnar epithelium (Fig 5A/1) with numerous goblet cells (Fig 5A/arrows). A brush border (Fig 5A'/white arrowhead) on the apical surface of the epithelial cells indicated the presence of microvilli. The lamina propria (Fig 5A/1') contained some mucous glands near the orifice of the stomach, and was underlaid with a longitudinal layer of the lamina muscularis (Fig 5A/1"). Both laminae showed black pigments (Fig 5A/*), the abundance of which decreased caudally. The degree of decrease was variable between individuals. The submucosal layer (Fig 5A/2) was composed of fine loose collagenous connective tissue containing blood vessels. The muscular tunic had an inner circular layer (Fig 5A/3) and an outer longitudinal layer (Fig 5A/3').

The **stomach** had a simple columnar epithelium (Fig 5B–5D/1). The surface showed areolae and foveolae. The basophile nuclei were positioned centrally, thus allowing the eosinophilic apical cytoplasm (Fig 5E–5F/black arrowhead) to be discerned from the basal cytoplasm, which was stained light-pink. In the cranial part of the stomach, the lamina propria (Fig 5B/1') contained serous glands (Fig 5B/a). At the body of the stomach, heterocrine gastric glands (Fig 5C/b) were abundant, which were composed of slightly basophilic chief cells (exocrinocytus

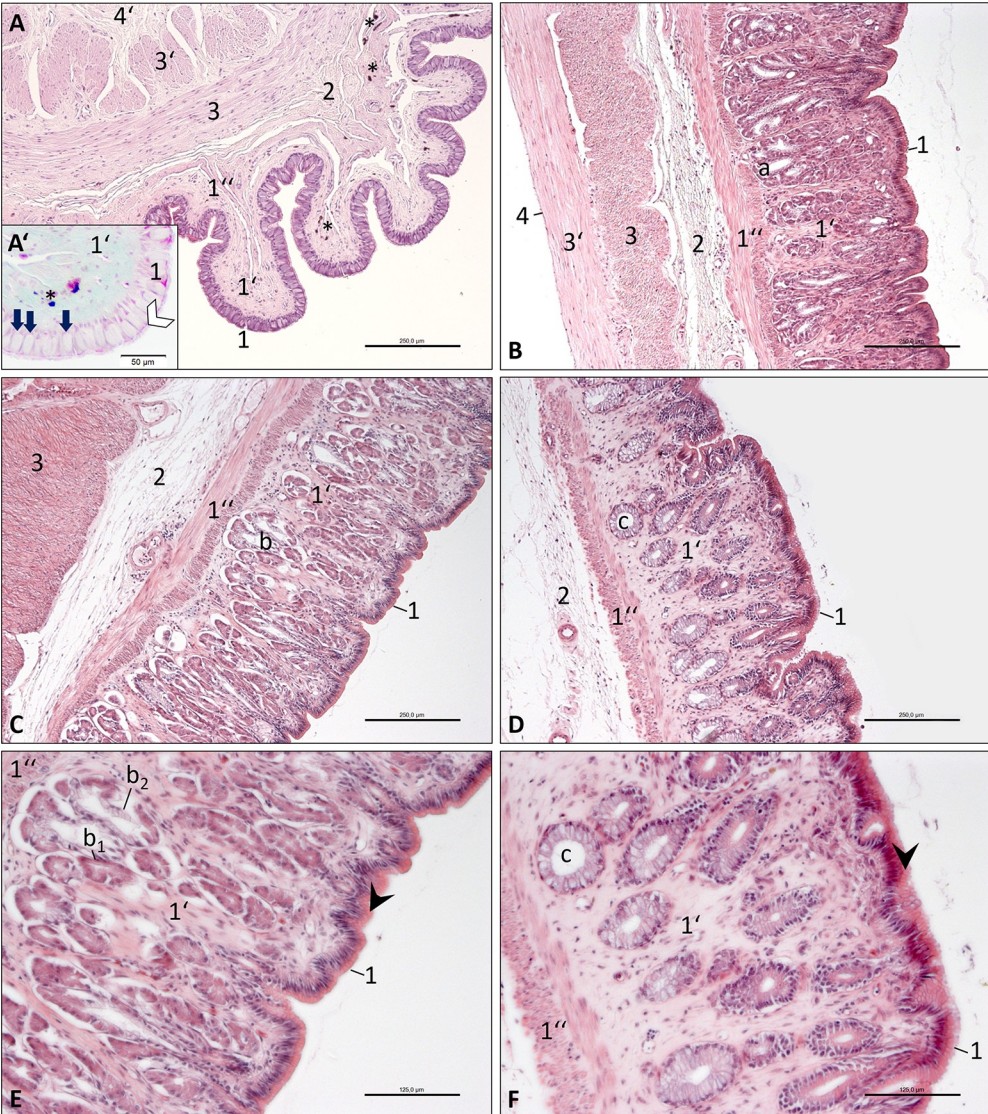

**Fig 5. Photomicrographs of the oesophagus and stomach wall, Inland Bearded Dragon (*Pogona vitticeps*); haematoxylin and eosin staining.** (A) oesophagus, transverse section, bar 250 μm; inset (A') Masson-Goldner staining, bar 50 μm; (B) cardiac part of stomach, longitudinal section, bar 250 μm; (C) body of stomach, longitudinal section, bar 250 μm; (D) pyloric part of stomach, transverse section, bar 250 μm; (E) body of stomach, longitudinal section, bar 125 μm; (F) pyloric part of stomach, transverse section, bar 125 μm. Mucous tunic consisting of epithelium (1), lamina propria (1') and muscular layer (1"); submucosal layer (2); muscular tunic consisting of an inner circular stratum (3) and an outer longitudinal stratum (3'); serosal tunic (4) or adventitia (4'). The lamina propria (1') contains serous glands (a) in the cardiac part (B) of the stomach; heterocrine glands (b) consisting of chief cells (exocrinocytus principalis; $b_1$) and acidophilic parietal cells (exocrinocytus parietalis; $b_2$) in the body (C, E); and mucous glands (c) in the pyloric part (D, F). The epithelium (1) of the oesophagus (A') shows a brush border (white arrowhead) and many goblet cells (arrows). The lamina propria (1') and the muscular layer (1") of the oesophagus (A, A') contain black pigments (*). E, F: The epithelial apical cytoplasm (1, black arrowhead) of the stomach appears eosinophilic.

principalis; Fig 5C/$b_1$) and acidophilic parietal cells (exocrinocytus parietalis; Fig 5C/$b_2$). Mucous glands (Fig 5D/c) were present in the distal, pyloric part of the stomach. The distinctly thick lamina muscularis of the mucosa had two layers (Fig 5B–5D/1"), an inner circular and an outer longitudinal layer. The submucosal layer consisted of delicate loose collagenous connective tissue with small arteries and veins (Fig 5B–5D/2). The muscular tunic was strong in

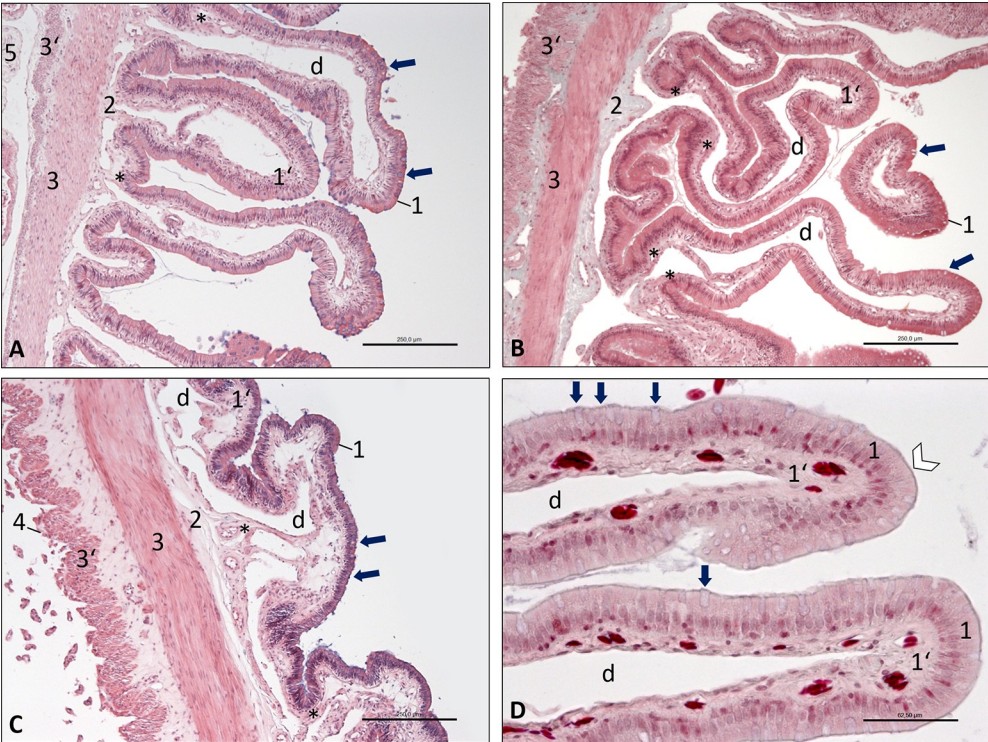

**Fig 6. Photomicrographs of the wall of the small intestine, Inland Bearded Dragon (*Pogona vitticeps*), transverse sections.** (A) duodenum, haematoxylin and eosin staining, bar 250 μm; (B) jejunum, Masson-Goldner staining, bar 250 μm; (C) ileum, haematoxylin and eosin staining, bar 250 μm; (D) duodenum, Masson-Goldner staining, bar 62.5 μm. Mucous tunic consisting of epithelium (1) interspersed with goblet cells (arrows) and lamina propria (1') with large central vessel (d); submucosal layer (2); muscular tunic consisting of an inner circular stratum (3) and an outer longitudinal stratum (3'); serosal tunic (4); attachment of mesentery (5). D: The epithelium (1) shows a brush border (white arrowhead). Smooth muscle cells (*) are located near the wall of the central vessel (d) in the lamina propria (1').

the stomach and consisted of two layers of smooth muscle cells, an inner circular and an outer longitudinal stratum (Fig 5B-C/3-3') in the proximal part. In the middle and distal parts, it had only a circular stratum (Fig 5C/3), the thickness of which increased remarkably at the distal part. The serosal tunic had a simple squamous epithelium with a faint lamina propria (Fig 5B/4), which was remarkably thicker in the middle and distal parts of the stomach. A clear border to a subserosal layer was not visible.

The **intestine** had a simple columnar epithelium (Figs 6A–6C/1, 5A–5D/1), which was interspersed with goblet cells (Figs 6A–6C/arrows, 5A-D/arrows), showing strong basophilic staining. The cytoplasm of the enterocytes was slightly acidophilic, and the nucleus was located in the basal half of the cells. The visibility of a brush border (Fig 6D/white arrowhead, Fig 7E and 7F/white arrowhead) indicated the presence of microvilli on the apical cell surface.

The small intestine (Fig 6A–6C) showed a large number of longitudinal mucosal folds decreasing distally in height and number. These folds contained a very large central vessel (Fig 6A–6C/d), which was surrounded by the lamina propria of the mucosa (Fig 6A–6C/1'). At the bases of the mucosal folds, the wall of these central vessels contained smooth muscle cells (Fig 6A–6C/*). The mucosa lacked a lamina muscularis and was separated from the muscular tunic only by a thin submucosal layer (Fig 6A–6C/2). The muscular tunic was comparatively thin, but consisted of two layers, an inner circular and an outer longitudinal stratum (Fig 6A–6C/3-

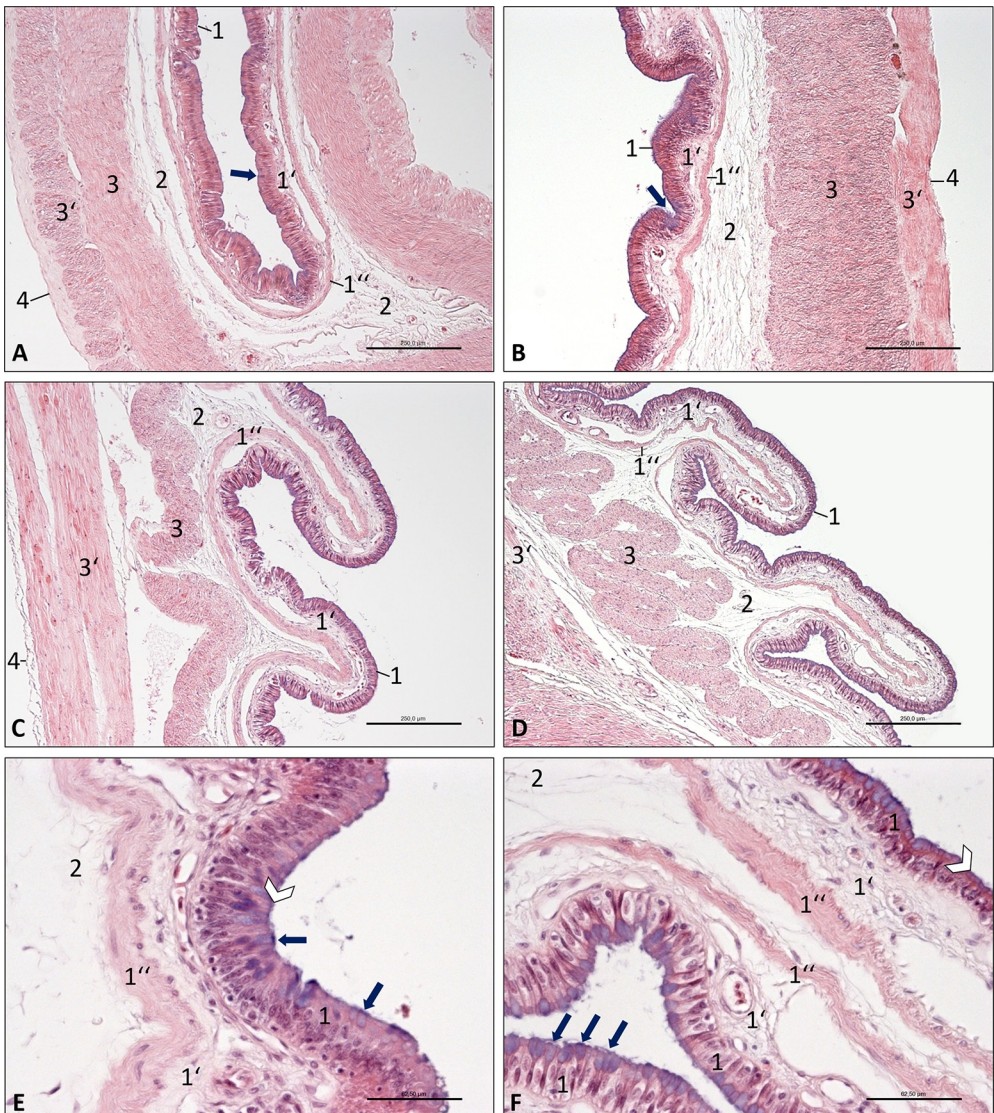

**Fig 7. Photomicrographs of the wall of the large intestine, Inland Bearded Dragon (*Pogona vitticeps*), haematoxylin and eosin staining.** (A) colic ampulla, bar 250 μm; (B) colic isthmus, bar 250 μm; (C) rectum, bar 250 μm; (D) cloaca, bar 125 μm; (E) colic ampulla; (F) cloaca, bar 125 μm. Mucous tunic consisting of epithelium (1) interspersed with goblet cells (arrows), lamina propria (1') and muscular layer (1"); submucosal layer (2); muscular tunic consisting of an inner circular stratum (3) and an outer longitudinal stratum (3'); serosal tunic (4). E, F: The epithelium (1) shows a brush border (white arrowhead).

3'). The serosal tunic was a simple squamous epithelium with a faint lamina propria (Fig 6A/4). However, the epithelial cells showed a cuboidal shape (Fig 6C/4) in some specimens.

The large intestine showed only a few mucosal folds in the colic ampulla (Fig 7A), while folds were more frequent (Fig 7B) in the colic isthmus. In the rectum and in the cloaca, distinct folds were present (Fig 7C and 7D). In comparison to the small intestine, all four parts of the large intestine lacked the large central vessel within the mucosal folds, but the mucosa had a clear, single-layered lamina muscularis (Fig 7A-7D/1"). This smooth muscle layer was divided from the muscular tunic by an obvious submucosal layer (Fig 7A-7D/2). The muscular tunic consisted of two strata: a circular and longitudinal one (Fig 7A–7D/3-3'). These two strata

were almost equally thick at the colic ampulla, while the thickness of the inner circular stratum increased remarkably at the colic isthmus, but was thinner again at the rectum. At the cloaca, the inner muscular stratum was relatively thin, but was arranged in loops. In comparison to the colon and rectum, the cloaca was no longer coated by a serosal tunic (Fig 7A–7C/4), but was surrounded by adventitia and perineal muscles.

## Discussion

The objective of the present study was to suggest a nomenclature for the different parts of the alimentary canal in bearded dragons, which is easily understandable and zoologically correct. As most veterinarian practitioners mainly treat mammals and the anatomical education at veterinarian schools is primarily based on domestic mammals, the findings of the present study were set in relation not only to other reptiles, but also to mammalian conditions. To meet this, the anatomical examination included the macroscopic anatomy, the topography of the alimentary canal itself and the neighbouring organs, as well as the blood vessel supply and the histology of the canal's wall. These morphological features allow the clear differentiation of the specific parts of the alimentary canal in mammals. However, there are distinct characteristics in the examined lizard species, which differ remarkably from the anatomy of mammals, but did in fact allow a clear distinction as well.

In bearded dragons, the **oesophagus** begins at the cranial aperture of the thoracic cavity. The transition from the pharynx to the oesophagus is perceptible because of the reduction in the lumen to a muscular tube and the loss of the pharyngeal dark pigmentation. This is true for many reptilian species [10]. Generally, the oesophagus is the organ that connects the pharynx to the stomach in vertebrates; it represents a muscular tube for the transport of food to the stomach [11–13]. In contrast to the bearded dragon, the mammalian oesophagus begins at the cranial end of the neck, passing along most of the neck and the entire length of the thoracic cavity [14–16]. The observed difference is due to the extremely short neck in the bearded dragon compared to the mammalian neck. In bearded dragons, the oesophageal epithelium is formed by a bilayered columnar epithelium with goblet cells. This feature is seen in most reptilian species [10], thus being in contrast to the stratified squamous epithelium of the mammalian oesophagus [14–16].

The macroscopic beginning of the **stomach** is unclear in bearded dragons because a diaphragm is missing and the diameter of the lumen only increases gradually. In contrast, the increase in the lumen of the mammalian stomach compared to that of the oesophagus is eminent and the cardia defines a clear border [14, 15]. In mammals, the cardiac notch separates the cardia from the blind-ending, dorsally bulging fundus on the left [15]. As this bulging part of the stomach is missing in bearded dragons, the cardiac part merges directly into the major part, the body of the stomach. However, in zoological textbooks [11–13] this major part of the stomach in amniotes is termed "fundus". In recent publications, the term "fundus" is used in one case for the major part of the stomach [3], and in another for the part caudal to the cardia [5]. Such inconsistencies may lead to misunderstandings. The terms "cardiac part" or "pars cardiaca" for the part of the stomach caudal to the cardia and "body of the stomach" or "corpus ventriculi" for the major part of the stomach, which extends until its turn to the right, are unequivocal. For the distal part of the stomach, the term "pyloric portion" [3], "pyloric antrum" [5] or "pyloric region" [12] can be found in literature. Thus, the terms are nearly the same as in textbooks on mammalian anatomy, in which the distal part of the stomach is called the pyloric part [14, 15] or pars pylorica [7, 16]. In some publications, the stomach is divided into descending, transverse and ascending parts in bearded dragons and other lizards [9, 17]. This does not apply to the topography of the stomach in live bearded dragons, which is

observed by radiographic studies [5, 6, 18]. Intra vitam, in this lizard species, the stomach runs only caudally, at first slightly to the left and then after turning, to the right. Thus, the terms "descending","transverse" and"ascending" may display the gastric topography in a specific reptile species, but are definitely not appropriate for bearded dragons.

To the best of the authors' knowledge, the present study describes the vascularisation of the alimentary canal of bearded dragons for the first time. In bearded dragons, the pattern of arterial blood supply changes between the cardiac part and the body of the stomach. The cardiac part of the stomach is provided with pairs of segmental arteries, which is the same pattern of blood supply as at the oesophagus. Thus, this feature is not valid as an indication of the location of the cardia. The major gastric arterial blood supply is accomplished by branches of the celiac artery, which is normally the second major artery leaving the aorta for the abdominal organs. Only in one examined bearded dragon was the celiac artery the first abdominal artery arising from the aorta, so that the order of aortic arteries in this specific animal more closely resembled that of mammals [14, 15]. In contrast to the arteries, the gastric veins drain the entire stomach to the portal vein, reaching even the cardiac part, but not the oesophagus. Thus, they can serve as an indication of the cranial border of the stomach.

In the bearded dragon, the microscopic structure of the epithelium changes gradually from the oesophagus to the stomach. The pseudostratified epithelium with goblet cells of the oesophagus with a moderate lamina propria is replaced by a simple columnar epithelium and a high lamina propria containing a great quantity of glands. In contrast to bearded dragons, a sudden change from oesophageal epithelium to gastric epithelium appears in humans and other mammals [19], and a strong caudal oesophageal [20] or cardiac musculature [16] prevents an oesophageal reflux of gastric content. However, this strong musculature is missing in the cardiac part of bearded dragons. This might explain why the oesophageal epithelium was often destroyed in the histologic specimens taken for the present study. The types of glands, which were found in the stomach of bearded dragons, are serous glands in the cardiac part, gastric glands in the body of the stomach and mucous glands in the pyloric part. A subdivision into these three glandular zones can also be observed in mammals [19, 21].

In contrast to the cardia, the pylorus is clearly defined by strong palpable musculature in bearded dragons. The pylorus points to the right side of the body cavity. Both features are similar to those of mammals [14–16].

In bearded dragons, the **small intestine** has a significantly smaller lumen than the large intestine, with the result that in this lizard species, the transition between both parts of the intestine is clearly defined as is true for most reptile [22] and mammalian species [11–13].

To divide the small intestine of bearded dragons into the three parts known in mammals [7, 14–16], several criteria have to be used. One criterion was the size of the intestinal lumen, which decreases from the duodenum towards the ileum. This is in accordance with radiographic results after administering a contrast medium in live bearded dragons [5]. However, different intestinal content influences the width of the lumen in cadaveric specimens. Thus, the narrowing of the lumen was not visible in all bearded dragons examined in the present study.

A further criterion is the opening of the bile duct into the cranial part of the small intestine, which marks this part as the duodenum. This feature can be observed in reptiles as well as in mammals [11–13].

The decisive criterion for the subdivision of the small intestine in bearded dragons is actually the blood supply. In bearded dragons, two major arteries arise close to the celiac artery as first and third ventral artery from the aorta and supply the intestine except its final caudal part. This is similar to many other reptile species, in which these two arteries were named the coecalic and the superior mesenteric artery [22, 23]. All three ventral arteries are formed by splitting

one common celiac-mesenteric artery, and their order occurs due to twisting of the resulting arteries [22, 23]. The first artery was termed the coecalic artery [22, 23], referring to the old name of the caecum. This artery actually supplies the ileum, the caecum and the colon. Therefore, the term "ileocolic artery", which is used in mammalian anatomical nomenclature, would display the real area of blood supply better. However, in the bearded dragon, this artery splits into two arteries, one leading to the ileum, caecum and colon, like in other reptiles, and the other to the duodenum. This splitting of the "coecalic artery" has been found in several other agamid lizards such as the Roughtail Rock Agama (*Stellagama stellio*, LINNAEUS, 1758) or the Dessert Agama (*Trapelus mutabilis*, MERREM, 1820), and can be seen in some gecko species such as the Sinai Fan-fingered Gecko (*Ptyodactylus guttatus*, HEYDEN 1827) and in some chameleon species such as the Graceful Chameleon (*Chamaeleo gracilis*, HALLOWELL, 1844), too [22]. Thus, even the term "ileocolic artery" is incomplete and "duodeno" should be added for bearded dragons. However, we suggest to keep using "ileocolic artery", as this term is appropriate for almost all reptilian species and not only for a few. Nevertheless, this particular arterial branch of the ileocolic artery to the duodenum enables a clear distinction between the duodenum and the remaining small intestine in bearded dragons. For the ileum, it is not the arterial supply, but the venous drainage, which divides it from the remaining small intestine. The ileal vein branches off the caudal mesenteric vein to drain the ileum. Unfortunately, in literature the venous distribution pattern in reptiles is not described in detail. Previous studies focused on the development and topography of the main venous stems [23, 24]. For the gastrointestinal tract, the main vein is the portal vein, which runs to the liver, collecting the venous blood from the common mesenteric vein. This is true not only for reptiles [23, 24], but also for birds [25, 26] and mammals [7, 14–16].

An additional criterion for differentiation could be the height of the mucosal folds of the small intestine, which decreases caudally. However, as it decreases continuously, it cannot serve as a distinct border between the different parts. The mucosal folds of the small intestine seen in bearded dragons are similar to those of several other lizard and reptile species [27]. On the contrary, the mammalian small intestine is equipped with intestinal villi [7, 14–16]. Histologic images of the study at hand reveal the presence of large vessels within the mucosal folds and at the base of the mucosal folds of the bearded dragon. This is in accordance with investigations concerning the lymphatic system in reptiles, which showed that the intestine of reptiles normally contains three nets of lymphatic vessels: subserous muscular, submucosal and mucosal ones [28]. In gecko and lacerta species, the mucosal network consists of longitudinal lymphatic vessels in the mucosal folds. The submucosal network is formed by longitudinal vessels, which are connected by large transverse anastomoses. In contrast to the small intestine, the principal lymphatic net of the large intestine lies in the submucosa where large vessels are arranged in large meshes [28]. These differences of the lymphatic networks might explain the observation that the wall of the small intestine, despite its thinness, shows up to five layers in the ultrasound image, while the layers of the wall of the large intestine cannot be differentiated in bearded dragons [4].

The intestinal epithelium of bearded dragons and other reptiles as well as that of mammals is similar, as all have columnar epithelial cells with microvilli interspersed with goblet cells [19, 21, 29].

A circular mucosal fold indicates the border between the **small** and **large intestine** in the bearded dragon. In amniotes, the ileocaecal valve is found in this specific location [11, 12]. However, the term is misleading as it connects the ileum with the colon [11, 12]. Due to this inconsistency, the term "ileocolic valve" is applied in specific descriptions of reptile anatomy [17, 27, 28]. The respective intestinal opening in domestic mammals is defined to be the ileal ostium/orifice [7, 14–16]. Some recent studies refer to the first part of the large intestine as

caecum in bearded dragons [3, 5] despite the fact that the caecum is defined as a diverticulum of the colon [12]. The caecum is positioned dorsally to the colon in reptiles [12] and it is therefore named "caecum dorsale" [30]. This is in accordance with the findings in most bearded dragons investigated for the present study, which displayed a small dorsal diverticulum protruding from the first part of the large intestine. This allows the conclusion that the diverticulum is the small, blind-ending caecum and the actual first part of the large intestine is the colon. As the first colic part is dilated to a nearly spherical shape in bearded dragons, the term "colic ampulla" was chosen in the present study. This term is also used for the extended final part of the equine ascending colon [7, 14, 16]. The following narrow segment separates the colic ampulla from the widened final part of the large intestine in bearded dragons. This narrow segment was termed the "colic isthmus" in the study at hand. As the final part of the intestine terminating in the cloaca is straight, the term "rectum" was chosen. This term is commonly used for the last straight part of the large intestine in birds [8, 31] and mammals [7, 14–16]. However, in zoological textbooks [11–13] and publications on reptiles [17, 27, 29], the colon is not subdivided into colon and rectum.

Looking at the arterial supply of the large intestine in bearded dragons, cranial arteries from the ileocolic artery and caudal arteries, namely the caudal mesenteric and the rectal arteries, are visible. The same is the case in birds and mammals with the limitation that the ileocolic (avian: ileocaecal) artery arises from the cranial mesenteric artery and the rectal arteries arise from the caudal mesenteric artery in birds [25, 26] and mammals [7, 14–16]. Additionally, in the examined bearded dragons, the ramification of the caudal mesenteric artery into a cranial and a caudal branch is similar to the distribution pattern in birds [25, 26] and mammals [7, 14–16]. In birds, the two branches are named "cranial branch" and "caudal branch" [25, 26], while in mammals the respective terms are "left colic artery" and "cranial rectal artery" [7, 14–16]. In the bearded dragon, two to three segmental arteries branch off the aorta caudal to the caudal mesenteric artery to supply the final part of the intestine. In the present study, these arteries were termed "rectal arteries", as they reach the intestine cranially to and directly at the border to the cloaca. This border is marked by at least one circular mucosal fold, and the cloaca is supplied by paired arteries arising from the iliac and renal arteries. These findings support the suggestion to name the final part of the large intestine "rectum".

The surface structure of the large intestine in bearded dragons, showing small protrusions all over its surface, is also seen in other reptile species. In the Jackson's chameleon (*Chamaeleon jacksonii*, BOULENGER 1896) villus-like projections can be found, and in some agamid and spiny-tailed lizards (*Uromastyx* species) as well as in some skinks (*Eumeces* species), a pebbled or papillose appearance of the luminal surface of the entire colon is observed [27]. This macroscopic finding is confirmed by histologic results in the present study, where small mucosal elevations are visible in longitudinal as well as in transverse sections, which proves that they cannot be folds. Apart from this, the epithelium remains similar to that of the small intestine, a feature that has been observed in different reptilian species previously [29]. In contrast, the mammalian large intestine lacks villous structures, and shows intestinal crypts [19, 21, 32]. The fact that such crypts cannot be found in bearded dragons corresponds to histologic findings in other reptiles [29].

In conclusion, the different parts of the alimentary canal of the inland bearded dragon display remarkable differences concerning the topography, anatomy and histology of the organ wall as well as vascular supply compared to mammals. Nevertheless, there are sufficient similarities to identify homologous parts and structures, suggesting a definite and understandable **nomenclature** for the parts of the alimentary canal and the supplying blood vessels in bearded dragons.

In the coelom of the bearded dragon, the alimentary canal begins with the **oesophagus** supplied by paired **oesophageal arteries**. The following **stomach** is subdivided into a **cardiac part**, a **body** and a **pyloric part**, characterised by their topography and type of glands, and supplied by paired **short gastric arteries** and branches of the **celiac artery**. Subsequently, the **small intestine** can be differentiated into a **duodenum**, a **jejunum** and an **ileum.** The jejunum is supplied by branches of the **cranial mesenteric artery**, while the duodenum and the ileum receive branches of the **ileocolic artery**. For the special characteristics of the **large intestine** in bearded dragons, the terms **colic ampulla** and **colic isthmus** were chosen; the small dorsal diverticulum of the colon is the **caecum** and the caudal continuation the **rectum**. Their arterial branches arise from the **ileocolic artery**, the **caudal mesenteric artery** and **rectal arteries**. The venous drainage is implemented by the **portal vein**, which receives a **gastric vein**, several small **duodenal veins** and a **common mesenteric vein**, into which a **cranial and caudal mesenteric vein** join.

## Acknowledgments

The technical assistance of Doris Voigtländer and Marion Langeheine preparing the specimens histologically is gratefully acknowledged. We most gratefully acknowledge the expert contribution of our native speaker, Ms. Frances Sherwood-Brock, who revised the English manuscript.

## Author Contributions

**Conceptualization:** Elisabeth Engelke, Christiane Pfarrer, Michael Fehr, Karina A. Mathes.

**Investigation:** Elisabeth Engelke, Katharina Radelof.

**Methodology:** Elisabeth Engelke, Christiane Pfarrer, Karina A. Mathes.

**Resources:** Katharina Radelof, Michael Fehr.

**Supervision:** Christiane Pfarrer, Michael Fehr, Karina A. Mathes.

**Validation:** Christiane Pfarrer, Karina A. Mathes.

**Visualization:** Elisabeth Engelke, Katharina Radelof.

**Writing – original draft:** Elisabeth Engelke.

**Writing – review & editing:** Christiane Pfarrer, Karina A. Mathes.

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
