## [Decision Letter · Decision Letter 0]

13 Mar 2020

PONE-D-19-31153

Gross anatomy, histology and blood vessel topography of the alimentary canal of the Inland Bearded Dragon (Pogona vitticeps)

PLOS ONE

Dear Dr. Engelke,

Thank you for submitting your manuscript to PLOS ONE. After careful consideration, we feel that it has merit but does not fully meet PLOS ONE’s publication criteria as it currently stands. Therefore, we invite you to submit a revised version of the manuscript that addresses the points raised during the review process.

We would appreciate receiving your revised manuscript by Apr 27 2020 11:59PM. To enhance the reproducibility of your results, we recommend that if applicable you deposit your laboratory protocols in protocols.io, where a protocol can be assigned its own identifier (DOI) such that it can be cited independently in the future. For instructions see: http://journals.plos.org/plosone/s/submission-guidelines#loc-laboratory-protocols

We look forward to receiving your revised manuscript.

Kind regards,

Jack Kottwitz

Academic Editor

PLOS ONE

Additional Editor Comments (if provided):

This was a well written manuscript. The authors have done a very good job. . I have no comments beyond what Reviewer 1 and Reviewer 2 noted in their respective reviews.

Journal Requirements:

2. Please clarify the source of the animal cadavers used in this study. If they were pets, please state whether you received permission from the owners. Please also provide a reference for the source if relevant.

"This publication was supported by Deutsche Forschungsgemeinschaft and University of Veterinary Medicine Hannover, Foundation within the funding programme Open Access Publishing."

4. Your ethics statement must appear in the Methods section of your manuscript. If your ethics statement is written in any section besides the Methods, please move it to the Methods section and delete it from any other section. Please also ensure that your ethics statement is included in your manuscript, as the ethics section of your online submission will not be published alongside your manuscript.

Reviewers' comments:

Reviewer's Responses to Questions

**Comments to the Author**

1. Is the manuscript technically sound, and do the data support the conclusions?

Reviewer #1: Yes

Reviewer #2: Yes

2. Has the statistical analysis been performed appropriately and rigorously? 

Reviewer #1: N/A

Reviewer #2: N/A

3. Have the authors made all data underlying the findings in their manuscript fully available?

Reviewer #1: Yes

Reviewer #2: Yes

4. Is the manuscript presented in an intelligible fashion and written in standard English?

Reviewer #1: Yes

Reviewer #2: Yes

5. Review Comments to the Author

Reviewer #1: The authors are to be commended for producing a well illustrated and well organized anatomical paper. The fact that multiple individual individuals of both sexes were used and that individual variations were noted is also to be commended. This paper contributes significant information about the anatomy of the blood supply in bearded dragons. Since these are some of the most common lizards in the captivity, this information is very useful. I have attached a copy of the manuscript with a few comments, including a few editorial suggestions and possible extra figures to include. I have also suggested a rewrite of the paragraphs describing veins to show their inherent difference from arteries. However, my overall impression is that this paper is well researched, well annotated and well written with little need for revision.

Reviewer #2: Line 108. Change the label of the testicle in the gross image. It is labeled “K”. I would switch the liver and testicle letters. Many practitioners easily forget that the kidneys in bearded dragons are within the pelvis. If they glance at the image, they will immediately think K=Kidney.

Line 139: If there are only two lobes of the pancreas, this should be noted.

Line 142: Is the spleen at the cranial end of the dorsal or ventral pancreatic lobe.

Line 150 to 157 and 167. Are there approximate measurements to these ‘larger, smaller, longer’ descriptions? I realize this might not be possible given the range in size of the females. The male specimens seem to be approximately the same size. Or maybe there is some way to give the reader a guesstimate: ‘the jejunum was about twice as long as the duodenum’. ‘The duodenum is ¼ larger in diameter than the jejunum.’ You do provide a measurement for the caecal diverticulum.

260 to 261: I want to make sure I understood this….the ‘smaller veins from the’ or the ‘further large vein left’ of the ‘cranial mesenteric vein’ drains the pancreas (both lobes?) and the spleen? I am unclear about ‘this branch’ arose form the caudal mesenteric vein (in two out of seven animals).

327 and 563: Ventriculus? Do bearded dragons have a ventriculus? Is there a reference for this? I thought this was only in crocodiles and alligators.

I really like the images!

6. PLOS authors have the option to publish the peer review history of their article (what does this mean?). If published, this will include your full peer review and any attached files.

Reviewer #1: Yes: Ray Wilhite

Reviewer #2: No

---

## [Author Response · Author response to Decision Letter 0]

29 Apr 2020

Response to reviewers 

Manuscript ID PONE-D-19-31153:

"Gross anatomy, histology and blood vessel topography of the alimentary canal of the Inland Bearded Dragon (Pogona vitticeps)"

We would like to thank the reviewers for their thorough proof-reading of the manuscript and their thoughtful comments. We have considered each comment carefully and revised our manuscript to address the issues raised. 

(The line numbers in our answers refer to the revised manuscript with track changes.)

Additional Editor Comments (if provided):

Journal Requirements:

Requirement #1. 

Please ensure that your manuscript meets PLOS ONE's style requirements, including those for file naming. The PLOS ONE style templates can be found at http://www.plosone.org/attachments/PLOSOne_formatting_sample_main_body.pdf and http://www.plosone.org/attachments/PLOSOne_formatting_sample_title_authors_affiliations.pdf

>>We inserted continuous line numbers. 

First page: We removed the “and” between the second last and last author, line 8. as the template for the first page does not show it.

File naming: We changed the captions of the figures composed of several images (Figs 2, 5, 6, 7) by removing the letters from the figure labels in the caption, so that the figure file names and the figure label in the caption are the same now. Fig 2: page 8, line 185; Fig 5: page 14, line 339; Fig 6: page 15, line 357; Fig 7: page 15, line 369.

Requirement #2. 

Please clarify the source of the animal cadavers used in this study. If they were pets, please state whether you received permission from the owners. Please also provide a reference for the source if relevant.

>>We inserted the information that we asked the owners of the pets for permission into the Material and method section of the manuscript, page 4, lines 72-73.

Requirement #3. 

Please remove any funding-related text from the manuscript…

>>We removed the funding information from the Acknowledgements section of the manuscript, page 27, lines 641-643.

… and let us know how you would like to update your Funding Statement. 

>>Please remove "The author(s) received no specific funding for this work." from the Funding Statement section and insert instead " This publication was supported by Deutsche Forschungsgemeinschaft (https://www.dfg.de) and University of Veterinary Medicine Hannover, Foundation (https://www.tiho-hannover.de) within the funding programme Open Access Publishing. The funders had no role in study design, data collection and analysis, decision to publish, or preparation of the manuscript."

Requirement #4. 

Your ethics statement must appear in the Methods section of your manuscript. If your ethics statement is written in any section besides the Methods, please move it to the Methods section and delete it from any other section. Please also ensure that your ethics statement is included in your manuscript, as the ethics section of your online submission will not be published alongside your manuscript.

>>We inserted our ethics statement into the Material and methods section of our manuscript, page 5, lines 99-103.

Reviewer #1:

Attached copy of the manuscript with comments

Comments:

Comment #1 

Page 3, line 50: Does this statistic apply to all lizards or just bearded dragons? 

>>The statistic applies to all examined lizards in the respective study [2].

Comment #2

Page 5, line 110: beautiful specimen preparations and i like the intuitive labels for the arteries.

>>Thank you! We enjoyed reading this comment!

Comment #3

Page 5, line 115: I would love to see a figure showing this.

>>As we did not receive an answer to our question to the editor about the permission to add figures until 24. April 2020, we did not add macroscopic figures.

Comment #4

Page 5, line 117: I think you mean "L", correct?

>>“L” is correct, we changed the letter, page 6, line 124. 

Comment #5

Page 6, line 131: I would suggest using "Where" instead of "When"

>>We followed the suggestion of the reviewer and inserted “Where” instead of “When”, page 6, line 138.

Comment #6 

Page 6, lines 133-136: I would love to see a photo of this included.

>>As we did not receive an answer to our question to the editor about the permission to add figures until 24. April 2020, we did not add macroscopic figures.

Comment #7

Page 8, lines 172-173: It would be good to have a figure showing the parts of the cloaca. I have seen very few good dissection images of the three parts.

>>As we did not receive an answer to our question to the editor about the permission to add figures until 24. April 2020, we did not add macroscopic figures.

Comment #8

Page 8, line 185: add "of stomach"

>>Following the Reviewer’s suggestion, we added “of stomach”, page 8, line 193.

Comment #9

Page 9, line 220: remove comma

>>We followed the suggestion of Reviewer 1, page 10, line 228.

Comment #10

Page 11, lines 249-270: I would strongly suggest rewriting this section on veins to reflect the fact that veins are named where they drain a target organ and then come together to form larger veins instead of branching to supply individual organs as arteries do. We try to stress this difference to the students and it would be good to see it emphasized in the literature.

>>According to the Reviewer’s suggestion, we removed the original text and re-wrote the description of the veins in the order of the blood flow, pages 11-13. 

While writing the new text, we recognized that the dorsal gastric veins would be described in a better way, when we add “dorsal” to their term. Therefore, we added “dorsal” to the labelling of these veins in Fig 4, page 14, 327-328.

Comment #11

Page 13, line 311-316: The histology figures are excellent. I would also like to see some higher res images of the smaller individual cells and structures discussed in the text if possible as well. I understand if this is not possible due to figure constraints, however.

>>Following the suggestion of the Reviewer, we inserted one higher-resolution image to Fig 6 and two higher-resolution images to Figs 5 and 7, respectively. We adapted the figure captions of the Figs 5, 6 and 7 to the changes made, pages 14-16. The figure references in the text were also adapted, page 16, line 379-390.

Inserting the higher-resolution images, we recognized that the clearly visible brush border of the oesophagus was not yet mentioned in the manuscript. Thus, we inserted a sentence describing the oesophageal brush border, page 16, lines 380-381.

Comment #12 and #13

Page 14, lines 328-329: delete add "to be discerned" here

>>We changed the sentence in the suggested manner, page 16, lines 389-390.

Comment #14

Page 14, line 347-348: Figure?

>>We added an indication of the brush border to Fig 6D, Fig 7E-F and a figure reference to the text, page 17, line 408-409.

Comment #15

Page 15, line 355: Should there be a figure reference here? Fig 6A-C/4-4' perhaps?

>>The Reviewer is right; there should have been a figure reference. We inserted the respective reference, page 17, lines 416-417.

Comment #16

Page 16, line 383: change to "as well"

>>We changed the wording in the suggested manner, page 18, line 444.

Comment #17

Page 22, line 544: add "also" here

>>We changed the wording in the suggested manner, page 24, line 605.

Reviewer #2:

Comment #1

Line 108. Change the label of the testicle in the gross image. It is labeled “K”. I would switch the liver and testicle letters. Many practitioners easily forget that the kidneys in bearded dragons are within the pelvis. If they glance at the image, they will immediately think K=Kidney.

>>We followed the suggestion of the reviewer and switched the letters indicating liver and testicle in Figure 1, the text (page 6, line 125) and the figure caption (page 5, line 115).

Comment #2

Line 139: If there are only two lobes of the pancreas, this should be noted.

>>There is also a pancreatic body between the two lobes. We inserted a sentence explaining this fact, page 7, lines 148-150.

Comment #3

Line 142: Is the spleen at the cranial end of the dorsal or ventral pancreatic lobe.

>>The spleen is located at the dorsal lobe of the pancreas. We added this detail to the respective sentence, page 7, lines 150-151.

Comment #4

Line 150 to 157 and 167. Are there approximate measurements to these ‘larger, smaller, longer’ descriptions? I realize this might not be possible given the range in size of the females. The male specimens seem to be approximately the same size. Or maybe there is some way to give the reader a guesstimate: ‘the jejunum was about twice as long as the duodenum’. ‘The duodenum is ¼ larger in diameter than the jejunum.’ You do provide a measurement for the caecal diverticulum.

>>Exact measurements of the specific sections of the alimentary canal were taken and have already been published by our work group in PLOS ONE [6]. To give the reader the information where to find the measurement values, we added an indication with the reference to Material and methods, page 4, lines 88-89.

Comment #5

260 to 261: I want to make sure I understood this….the ‘smaller veins from the’ or the ‘further large vein left’ of the ‘cranial mesenteric vein’ drains the pancreas (both lobes?) and the spleen? I am unclear about ‘this branch’ arose form the caudal mesenteric vein (in two out of seven animals).

>>We re-wrote the chapter about the veins according the suggestion of reviewer 1. In doing so, we explained that it is the left dorsal gastric vein draining the pancreas and spleen, page 13, lines 309-311.

We added “the dorsal lobe of” to “the pancreas”. page 13, line 310

Comment #6

327 and 563: Ventriculus? Do bearded dragons have a ventriculus? Is there a reference for this? I thought this was only in crocodiles and alligators.

>>Checking the text of our manuscript, we became aware of the fact that we did not use any other Latin term for other organs or stuctures. Thus, we removed the Latin term “ventriculus” from the text, page 16, line 388 and page 25, line 625.

The term “ventriculus” is the Latin anatomic term for the single-chambered stomach in veterinary medicine [7]. In birds and crocodiles/alligators, there is an additional gastric chamber, the proventriculus, which is located cranially to the “main” stomach, the ventriculus. In birds, the proventriculus is also called glandular stomach, and the ventriculus is also known as muscular stomach or gizzard.

Again, the authors are grateful for the very helpful editorial suggestions and improving remarks.

---

## [Decision Letter · Decision Letter 1]

2 Jun 2020

Gross anatomy, histology and blood vessel topography of the alimentary canal of the Inland Bearded Dragon (Pogona vitticeps)

PONE-D-19-31153R1

Dear Dr. Engelke,

We are pleased to inform you that your manuscript has been judged scientifically suitable for publication and will be formally accepted for publication once it complies with all outstanding technical requirements.

With kind regards,

Jack Kottwitz

Academic Editor

PLOS ONE

Reviewers' comments:

Reviewer's Responses to Questions

**Comments to the Author**

1. If the authors have adequately addressed your comments raised in a previous round of review and you feel that this manuscript is now acceptable for publication, you may indicate that here to bypass the “Comments to the Author” section, enter your conflict of interest statement in the “Confidential to Editor” section, and submit your "Accept" recommendation.

Reviewer #1: All comments have been addressed

Reviewer #2: All comments have been addressed

2. Is the manuscript technically sound, and do the data support the conclusions?

Reviewer #1: Yes

Reviewer #2: Yes

3. Has the statistical analysis been performed appropriately and rigorously? 

Reviewer #1: N/A

Reviewer #2: N/A

4. Have the authors made all data underlying the findings in their manuscript fully available?

Reviewer #1: Yes

Reviewer #2: Yes

5. Is the manuscript presented in an intelligible fashion and written in standard English?

Reviewer #1: Yes

Reviewer #2: Yes

6. Review Comments to the Author

Reviewer #1: I very much appreciate the author's efforts at addressing the reviewer comments. The vein section, in particular, is greatly improved by the resdescription.

Reviewer #2: (No Response)

7. PLOS authors have the option to publish the peer review history of their article (what does this mean?). If published, this will include your full peer review and any attached files.

Reviewer #1: Yes: Ray Wilhite

Reviewer #2: No

---

## [Editor Report · Acceptance letter]

8 Jun 2020

PONE-D-19-31153R1 

Gross anatomy, histology and blood vessel topography of the alimentary canal of the Inland Bearded Dragon (Pogona vitticeps) 

Dear Dr. Engelke:

I'm pleased to inform you that your manuscript has been deemed suitable for publication in PLOS ONE. Congratulations! Your manuscript is now with our production department. 

Kind regards, 

on behalf of

Dr. Jack Kottwitz 

Academic Editor

PLOS ONE